# Recycling of Wind Turbine Blades into Microfiber Using Plasma Technology

**DOI:** 10.3390/ma16083089

**Published:** 2023-04-13

**Authors:** Žydrūnas Kavaliauskas, Romualdas Kėželis, Viktorija Grigaitienė, Liutauras Marcinauskas, Mindaugas Milieška, Vitas Valinčius, Rolandas Uscila, Vilma Snapkauskienė, Dovilė Gimžauskaitė, Arūnas Baltušnikas

**Affiliations:** Plasma Processing Laboratory, Lithuanian Energy Institute, Breslaujos Str. 3, LT-44403 Kaunas, Lithuania

**Keywords:** plasma, fiber, waste recycling, wind turbines, energy

## Abstract

As the industry develops and energy demand increases, wind turbines are increasingly being used to generate electricity, resulting in an increasing number of obsolete turbine blades that need to be properly recycled or used as a secondary raw material in other industries. The authors of this work propose an innovative technology not yet studied in the literature, where the wind turbine blades are mechanically shredded and micrometric fibers are formed from the obtained powder using plasma technologies. As shown by SEM and EDS studies, the powder is composed of irregularly shaped microgranules and the carbon content in the obtained fiber is lower by up to seven times compared with the original powder. Meanwhile, the chromatographic studies show that no hazardous to the environment gases are formed during the fiber production. It is worth mentioning that this fiber formation technology can be one of the additional methods for recycling wind turbine blades, and the obtained fiber can be used as a secondary raw material in the production of catalysts, construction materials, etc.

## 1. Introduction

In recent years, global industry and technologies have been evolving at a very rapid pace, resulting in ever-increasing energy needs. Today, the global industry derives most of its energy from fossil fuels in addition to other polluting technologies, but a rapidly growing share of energy comes from renewable energy sources such as wind and solar power plants, hydrogen, and others. According to officially available data, in the last 25 years, the amount of electricity and wind power plants increased up to 80 times [1,2,3,4,5,6]. Most wind plants are built far from the city in open areas, thus avoiding noise and obstacles that can naturally reduce the speed of the wind flow. Wind energy is clean and environmentally friendly as no gas or other environmentally hazardous substances are emitted during the electricity generation [7,8,9,10]. The price of wind turbines is determined by their structural features. It is financially worthwhile to build wind turbines with a power of more than 100 kW [11,12,13,14].

Fiberglass-reinforced plastic is used for the production of wind farms. By using composites of such construction, the required mechanical properties of wind turbine blades, such as flexibility, strength, and lightness, are achieved. In addition, it is worth mentioning that these composites are significantly more resistant to mechanical fatigue than metals, e.g., steel. Epoxy resin (C_21_H_25_ClO_5_) and polyester are also used in the production of composite materials for the production of wind turbines [15,16,17,18]. For turbine blades longer than 50 m, carbon fiber is used as an ingredient, which gives the design frame strength and lightness. In the production of wind turbine blades, the aim is to ensure that the composites used have the longest possible service life. In recent years, there has been increasing focus from researchers and industry on the circular economy and recycling of waste. Wind energy, which generates huge amounts of obsolete turbine parts and blades, is no exception. Scientists predict that in the period from 2022 to 2034 [19,20,21,22,23], 200,000 tons of rotor blade waste from installed wind turbines that have reached their standard service life will be generated. All of this waste can be recycled and further used in industries such as automotive, electrical and electronic equipment, and others. The recycling of wind turbine blades has some problems due to the composite composition as it is difficult to apply melting technology. Two types of technology, mechanical and thermochemical, are commonly used to recycle waste of this origin [1,24,25,26].

During mechanical processing, the blades of wind generators are crushed or ground and the resulting materials are then classified as a powder. The obtained materials can be used as a filler or reinforcement by re-impregnating them with a resin and they can also be added as an ingredient to asphalt or cement. However, the use of these recycled materials poses certain technological problems, such as changes in the thermal conductivity and viscosity of the new cement mix and the occurrence of side chemical reactions between the individual components of the mix [1,2,27].

Thermochemical methods have been used to obtain raw materials that have similar mechanical properties to those prior to their initial use, although surface defects and a decrease in mechanical resistance are common. The pyrolysis method can be used to recycle not solidified resins, and it is also possible to recycle carbon fiber while maintaining its original morphology. Pyrolysis is one of the most common recycling technologies as it is based on the thermal decomposition of a polymer in an inert environment (e.g., nitrogen). In this process, the organic portion of the composite decomposes into a molecular weight component that could in some cases be recovered and reused [3,4,28]. In the scientific literature, there are cases described where, by means of pyrolysis processing of wind turbine blades, an attempt was made to extract synthetic gases such as hydrogen and methane, but after conducting an economic evaluation of this process, it turned out to be financially unprofitable [1,29]. One of the thermal processing processes is oxidation. This process consists of a thermal decomposition of the polymer matrix at a temperature of 500 °C in a high oxygen flow. Using a cyclone separator, the fibers can be separated and then the remaining compounds can be fully oxidized in a secondary burner [15,30,31].

One of the most advanced and new methods of recycling wind turbine blades is the plasma method, with temperatures of 15,000 °C and above. During this process, all hazardous compounds are completely decomposed [21]. This method can be used to obtain micrometric fibers and melt. Various polymeric composites can be reinforced with obtained fibers in the production of construction materials due to the specific mechanical properties of the fiber (such as flexibility and plasticity). Fiber is increasingly used in construction, automotive, space, and energy industries. The fiber also can be used to produce gas catalysts for various purposes. The mineral fiber production process developed at the Lithuanian Energy Institute using plasma-chemical reactors provides an opportunity to form fine fibers from any secondary materials, including shredded wind turbine blades, using the kinetic energy of the plasma flow. Such a microfiber (in some cases up to 300 nm thick) can be used as a catalyst support for cheaper catalytically active metals such as Cu and TiO_2_, because its melting point is higher than the operating temperatures of the catalysts. The use of plasma technology for wind turbine processing and fiber formation has not been studied in the scientific literature, so this work is completely new and original.

The aim of this work is to use a plasma process to obtain a fiber suitable for the production of composite materials from shredded wind turbine blades and to study its properties and evaluate the environmental aspects of the plasma processing process using various methods.

## 2. Materials and Methods

Plasma technology was used to recycle the wind turbine blades. The primary feedstock that was injected into the plasma fiber formation reactor was obtained by mechanical comminution of the wind turbine blades. A special mill ML-1 was used to shred these blades. The primary raw material was comminuted to a powder with a diameter of up to 150 µm.

A plasma generator with an electric current of 178 A and a power of 68 kW was used to form the plasma. The plasma-forming gas was air with a total flow rate of 13 g/s and the ionized gas flow speed reached 1000 m/s at the outlet nozzle of the reactor. The temperature of the plasma flow at the exit of the reactor was about 2300 °C. In this work, the tested fiber was obtained using a special plasma-chemical reactor, the scheme of which is shown in Figure 1.

A special micrometric wire mesh was used to collect the resulting microfibers with a hole size of 200 µm. A Hitachi S-3400 N model scanning electron microscope (SEM) was used to evaluate the morphology of the primary powders and the thickness and geometrical characteristics of the obtained fiber. The energy dispersive X-ray spectroscopy (EDS) method was used to evaluate the elemental composition of the microfibers and primary powders, which was performed simultaneously with surface scanning electron microscopy. X-ray diffraction (XRD) with standard Bragg–Brentan focusing geometry was used to investigate the crystalline structure of the primary powders and microfibers. A TIGER S 8 x-ray fluorescence spectrometer (XRF) was used to determine the chemical composition of the original powder and the resulting fiber. Thermogravimeter Netzsch STA 449 F3 (TG) was used for the thermal analysis of the primary powders and fibers. The analysis of gas samples was performed using an Agilent 7890A gas chromatograph, when gas samples were previously taken during the formation of the fiber plasma process in special plastic containers by taking gas samples with the help of a metal water-cooled probe. The Fourier-transform infrared spectroscopy (FTIR) was used for the evaluation of functional groups in the feedstock powder and microfiber. By analyzing the FTIR spectra, the changes in the characteristic peaks of the organic compounds were evaluated, revealing the dynamics of substances of an organic origin occurring during fiber formation. The spectra were obtained using ALPHA FTIR spectrometer, Platinum diamond ATR module (Bruker Optic GmbH, Billerica, MA, USA) at room temperature in the range of 400–4000 cm^−1^ with a spectral resolution of 4 cm^−1^ and 30 scans were performed. Background spectra were measured before each measurement and background subtraction was done using software (OPUS, Bruker Optik GmbH, Billerica, MA, USA). The BET method was used to determine the specific surface area and porosity of the primary powders and the obtained microfibers. Before BET tests, the samples were degassed at 200 °C.

## 3. Results and Discussion

Figure 2 shows the morphology of the primary powder granules and the fiber obtained using plasma technology. The distribution of the elemental composition of the material in both the primary powder and the fiber is shown using EDS methodology. In Figure 2a we can see the surface morphology and granule size of the primary powder from which the fiber was formed. As shown by the analysis of SEM images, the primary powder consisted of irregularly shaped granules with geometric dimensions of up to 200 µm, and elongated rod-shaped derivatives could be detected in the primary powder. Figure 2b shows the fiber obtained from the primary powder. The fiber was obtained by injecting the powder into the air plasma flow. As it can be seen from the SEM image, the fiber is composed of elongated filaments of various thicknesses of up to 40–50 µm in diameter. The filaments are intertwined in a chaotic manner, forming structures resembling wool. Figure 2c shows the distribution of the elemental composition of the primary powder. As we can see, the powder consisted of Al, O_2_, C, Ca, Si, S, Na, and Mg elements. The colored map of the distribution of elements showed that the primary powder contained mostly carbon and oxygen.

It is likely that the amount of individual oxygen as an element was very small and most was in compounds with other elements. Figure 2c shows the color distribution of the elemental composition of the fiber. In this case, the largest amounts were of oxygen, silicon, and calcium. As we can see, there were significantly fewer carbon derivatives compared with the primary powder. This result can be explained by the fact that most of the carbon reacted during the fiber production process, turning into volatile compounds.

Table 1 shows the elemental composition studies using EDS for the primary powder and the obtained fiber. As we can see from the research, the primary powder consisted of elements, with the highest percentage being C, O_2_, Ca, Si, and Al. Other elements such as Si, Al, and Na made up a negligible percentage. In terms of atomic mass units, carbon and oxygen accounted for the largest share, of 58% and 35.4%, respectively; Ca, Si, and Al accounted for 1.5%, 3.2%, and 1.3%, respectively. It is likely that oxygen mostly existed in the form of oxides to form individual chemical compounds. The table also shows the percentage of elements, which differed significantly for some elements from the composition given using atomic mass (e.g., Ca—4%, Si—6.2%) units. Table 1 also shows the EDS elemental composition studies of the fibers obtained from the primary powder. In this case, the predominant elements with the highest concentration in atomic mass units were C—8.2, O_2_—56.5, Ca—6.7, Si—15.5, and Al—9.7%. Oxygen, as in the case of primary powders, existed in the form of individual compounds. Meanwhile, other elements such as Na, Ti, and Mg made up a significantly small fraction. As we can see, additional elements such as F, Ni, and Fe, were formed during the fiber formation process, but their total composition by atomic mass was less than 1%. These additional elements were most likely to occur during fiber formation, as the fiber was formed using air plasma and a plasma generator. The parts of the plasma generator were made of various types of metals such as Ni and Fe, so it is likely that at high plasma temperatures, because of erosion, a small portion of these materials entered the overall fiber composition.

Table 1 also shows the percentage elemental composition of the fiber, which, for some elements, differed significantly from the composition given using the atomic mass units (e.g., C—4.7% Ca—14.3%, and Si—20.9%). An analysis of the EDS results shows that the carbon content was significantly higher (about seven times) in the primary powder than in the formed fiber. According to the literature, large amounts of epoxy resin C_21_H_25_ClO_5_, for which one of the components is carbon, are used in the production of wind turbine blades. As the fiber formation takes place using air plasma and the process temperature reaches about 2300 K, carbon is converted into volatile compounds and its content is significantly reduced [1].

X-ray fluorescence (XRF) was used to determine the chemical composition of both the primary powder and the obtained fiber. The results of the XRF studies are presented in Table 2. As we can see, the highest percentage in the primary powder was SiO_2_—15.4, CaO—10.7, and Al_2_O_3_—3.38%; other compounds such as ZrO_2_, P_2_O_5_, and MgO made up a negligible proportion of the total composition. Analyzing the composition of the fiber compounds, it can be seen that it differed significantly from the composition of the primary powder. In this case, the predominant elements were SiO_2_—36.4, CaO—18.5, and Al_2_O_3_—19.9%. The main disadvantage of the XRF method is that it is not possible to capture elements and compounds lighter than carbon, so the overall percentage was not equal to 100%. According to the fact that the SiO_2_ content in the fiber increased up to 36.4%, (which corresponds to about 2.4 times), CaO increased by about 1.7 times, and Al_2_O_3_ increased by about 6 times, it can be concluded that the total carbon content drastically decreased, which was also confirmed by the EDS studies.

During fiber formation, the concentration of the formed gas was determined using chromatography. Plasma samples were collected from the plasma reactor using special bags and a water-cooled probe. The results of the measurements showed that the total gas concentration was O_2_—17.8, N_2_—82.5, CH_4_—0, CO_2_—2.2, CO—0, C_2_H_6_—0, C_2_H_2_—0, C_3_H_8_—0, and H_2_—0%. It is apparent that the amount of nitrogen and oxygen was close to the amount of these gases naturally in the atmosphere. During the fiber formation process, a small amount of CO_2_ was generated—2.2%. No harmful volatile compounds were formed during the process, so the additional purification of gases before their release into the environment or their use in other processes was not necessary.

Figure 3 presents the results of the thermogravimetry of primary powders and fibers using a reactive air gas environment. These studies were performed by raising the temperature from 0 to 900 °C. Figure 3a presents the thermogravimetric studies of the primary powders.

As it can be seen from the results of the measurement, the main process of weight loss of the material took place in the temperature range of approximately 300–500 °C. In this temperature range, the total weight of the primary powder decreased by about three times and reached about 30% of the weight of the starting material. This process is associated with a reduction in carbon content with an increase in temperature. This result correlated well with the EDS studies when the carbon content of the primary powder was about seven times higher than that of the fiber. The DTG curve shows that the most intense material loss process occurred at 400 °C when the material loss rate was about −28%/min. Figure 3b shows the results of the thermogravimetric study of the fiber. As the temperature was raised from 0 to 900 °C, the total mass of the material varied slightly—it decreased by about 1.5%. This result was because the fiber was free of volatile compounds and the carbon content was reduced up to seven times compared with the original powder.

Figure 4 presents the XRD studies of the primary powder and obtained fiber. As the analysis of the results shows, both the primary powder and the formed fiber had amorphous structures. Crystallization processes were not observed during the fiber formation process in the plasma environment. Amorphous fiber is more suitable for thermal insulation or for the production of catalysts because it has better insulating properties than materials with a crystalline structure.

The FTIR transmittance spectrum of the powder was characterized by a wide O-H band at 3440 cm^−1^ and an intense C-H band at ~2920 cm^−1^ in range of 2800–3100 cm^−1^, which are typical stretching vibrations of CH, CH_2_ and CH_3_ groups. The high intensity and narrow low intensity absorption peaks at 1720 cm^−1^ and 1600 cm^−1^ are assigned to the C=O and C=C groups, respectively. The peak at ~1443 cm^−1^ is due to C-C stretching vibrations in aromatic rings or C-H group vibrations. The absorption bands at 1061 cm^−1^ and 1248 cm^−1^ are attributed to the C-O stretching vibrations. The peak at ~1120 cm^−1^ is related to the vibrations of Si-O-C and/or C-O-C bonds. The sharp peaks at 740 cm^−1^ and 700 cm^−1^ are assigned to C-H out of plane of aromatic rings [28,29,30,31]. The existence of these characteristic absorption peaks was caused because the powder contained a large amount of epoxy resin, which is used to attach individual parts of the wind turbine blades—a significant part of which is of an organic origin [29,30,31]. The EDS measurements indicated the presence of low amount of Si, Ca and Al in the powders. Meanwhile, the SiO_2_, CaO and Al_2_O_3_ compounds were determined by XRF technique. However, it was hard to identify silicon oxide, calcium oxide or aluminum oxide functional groups in the FTIR spectrum due to the dominating epoxy resin vibrations groups and the low content of these elements in feedstock powder. 

Figure 5b shows the FTIR spectrum of the produced fiber. Several broad and high intensity absorption peaks were obtained in the wavenumbers range of 400–1250 cm^−1^. It was demonstrated that the stretching vibration of Al-O bonds are obtained at 850 cm^−1^ and 690 cm^−1^ [29,31]. The Si-O stretching vibrations appear at ~450 cm^−1^ (Si-O-Si), 970 cm^−1^ (Si-O-Si) and 1120 cm^−1^ (Si-O-Si) [29,30,31]. The calcium oxide vibration bonds usually are found at 970 cm^−1^ and 1020 cm^−1^ [30,31]. The existence of a broad peak in the range of 550–1250 cm^−1^ indicated that the silicon oxide, calcium oxide and aluminum oxide bands were overlapped. Meanwhile, the low intensity peaks observed at 1720 cm^−1^, 2850 cm^−1^ and 2920 cm^−1^ are attributed to the C=O bond, symmetrical C-H stretching of –CH_3_ bonds and asymmetrical C-H stretching of –CH_2_ group, respectively (Figure 5b). The wide absorption peak at ~1400 cm^−1^ is assigned to C=O bending vibrations. Such dynamics for the carbon containing functional peaks in FTIR spectra were obtained because during the formation of fibers a high temperature was used (up to 2300 °C) and the most organic compounds were decomposed, converted to volatile compounds or reduced to negligible amounts; thus, the intensities of the O-H, C-H, C-O, C=O or C=C absorptions peaks attributed to epoxy resin material were drastically reduced or not observed at all. The EDS measurements indicated that the concentration of the carbon was only 4.7 wt.% in the produced fiber. Meanwhile, the fiber consisted of the mixture of SiO_2_, CaO and Al_2_O_3_ [28,29,30,31]. 

Figure 6 presents the results of the BET measurements of the primary powder. As the analysis of the results shows, the specific surface area of the powder was 1.25 m/g^2^. The best suited calculation model for the evaluation of the pore size distribution in a specimen was QSDFT, N2, carbon equilibrium transition kernel at 77.4 K based on a slit-pore model, with a calculation fitting error equal to 3.270%. The largest total surface area was occupied by pores with a diameter of 1.83 nm. It is apparent that the predominant range in which the pores were distributed was from 1 to 10 nm. It is worth mentioning that the characteristics of the porous structure depended on the method of preparation of the primary powder. It also depended on the fraction obtained from the powder and the geometric shape of the granules.

Figure 7 presents the results of BET studies of the obtained fibers. The studies were performed under the same conditions as for the primary powder. In this case, the specific surface area was 1.04 m/g^2^. The most common pore width value was equal to 2 nm. It can be observed that the predominant range in which the diameter of the pores were distributed was 1–20 nm. It can be stated that the process of fiber formation had a small influence on the changes of the specific surface area and the porosity structure.

It is worth mentioning that the porosity and specific surface area of the obtained fiber depended on formation conditions such as the plasma flow velocity, temperature when leaving the reactor gas, and chemical composition of the primary powder. Although the specific surface area of the obtained fiber was not large, the fiber of such a structure could be successfully used in the production of catalysts for various purposes or as a thermal insulation layer. The additional injection of copper, magnesium, titanium, or other catalytic particles into the plasma flow during the fiber formation process could provide a composite for use in catalytic processes.

## 4. Conclusions

Micrometric size fibers were formed using plasma technology as the powder were obtained from wind turbine blades. A special cooling reactor was designed and fabricated to form the fiber from the primary powder, which was fed into the plasma flow from the external source. As shown by SEM studies, the primary powders obtained by mechanical comminution of turbine blades are composed of irregularly shaped granules or rod-shaped microforms. The EDS studies indicated that the primary powder consisted of elements, with the highest percentage being C, O_2_, Ca, Si, and Al. Meanwhile, the analysis of the EDS results of fibers showed that the carbon concentration is reduced by seven times. Chromatographic studies of the gases obtained during fiber formation showed that the predominant gas in the plasma flow was nitrogen and oxygen, and 2.2% of CO_2_ was also formed, which means that no environmentally hazardous gases were formed during the fiber formation process and thus it did not require further purification. XRF analysis of the composition of the fiber compounds shows that it differed significantly from the composition of the primary powder. In this case, the predominant elements were SiO_2_—36.4%, CaO—18.5 %, and Al_2_O_3_—19.9%. Thermogravimetric studies show that when the temperature was raised to 900 °C, the primary powder lost about 70% of its original weight. The most active weight loss process took place at 400 °C when the rate of material loss was about −28%/min. Meanwhile, the analysis of the fiber thermogravimetry results showed that in this case, the weight loss was very small and varied by about 1.5%. Both the primary powder and the formed fiber had amorphous structures. The FTIR transmittance spectrum of feedstock powder demonstrated characteristic peaks of the C-O, C-H_x(x=1,2,3)_, C=O, C=C, and OH groups, the existence of which was due to the use of epoxy resin in the production of wind turbine blades. Meanwhile, only low-intensity C-H and C=O bands were obtained in the FTIR spectrum of the fiber. The dominant absorption bands in the fiber were attributed to Al-O, Si-O and Ca-O groups. This means that most of the epoxy resin was successfully utilized during fiber production and the obtained fiber was an environmentally friendly raw material. BET studies show that the specific surface area of both the primary powder and the fiber was small, reaching 1.25 and 1.04 m/g^2^, respectively. Meanwhile, porosity studies show that both the surface of the powder and the fiber consisted of micropores with a diameter of less than 2 nm. The summarized research results show that the fiber formed from the powder obtained from the crushed wind turbine blades was sufficiently clean and environmentally friendly, so it could be widely used as a raw material for the production of thermal insulation, in the production of catalysts, or as a component in concrete, cement, or asphalt and in other industries.

## Figures and Tables

**Figure 1 materials-16-03089-f001:**
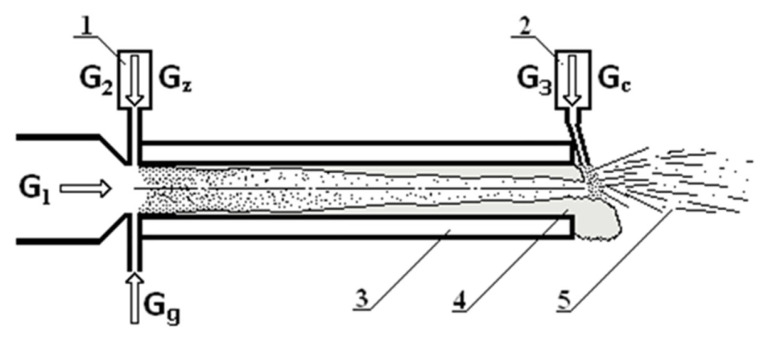
The schematic view of the catalytic fibre production process. 1—the feeder of particles; 2—the feeder of catalytic particles; 3—reactor walls cooled with water; 4—melt flow; 5—produced fibers and granules; G1—air plasma flow from plasma torch; G2—air flow for injection of ceramic materials; G3—air flow for spraying of catalytic particles; Gc—flow of catalytic particles; Gg—propane gas flow; Gz—flow of particles.

**Figure 2 materials-16-03089-f002:**
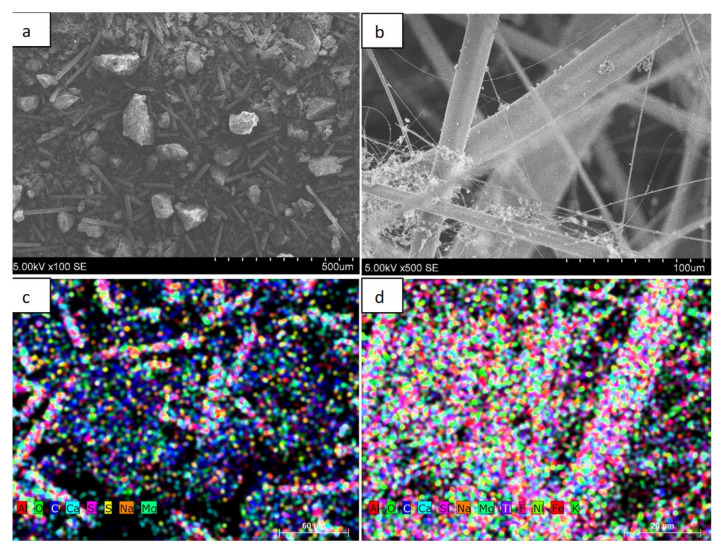
Images of wind turbine blade powder and fiber and distribution of elemental composition: (**a**) powder; (**b**) fiber; (**c**) distribution of the elemental composition of the powder; (**d**) distribution of the elemental composition of the fiber.

**Figure 3 materials-16-03089-f003:**
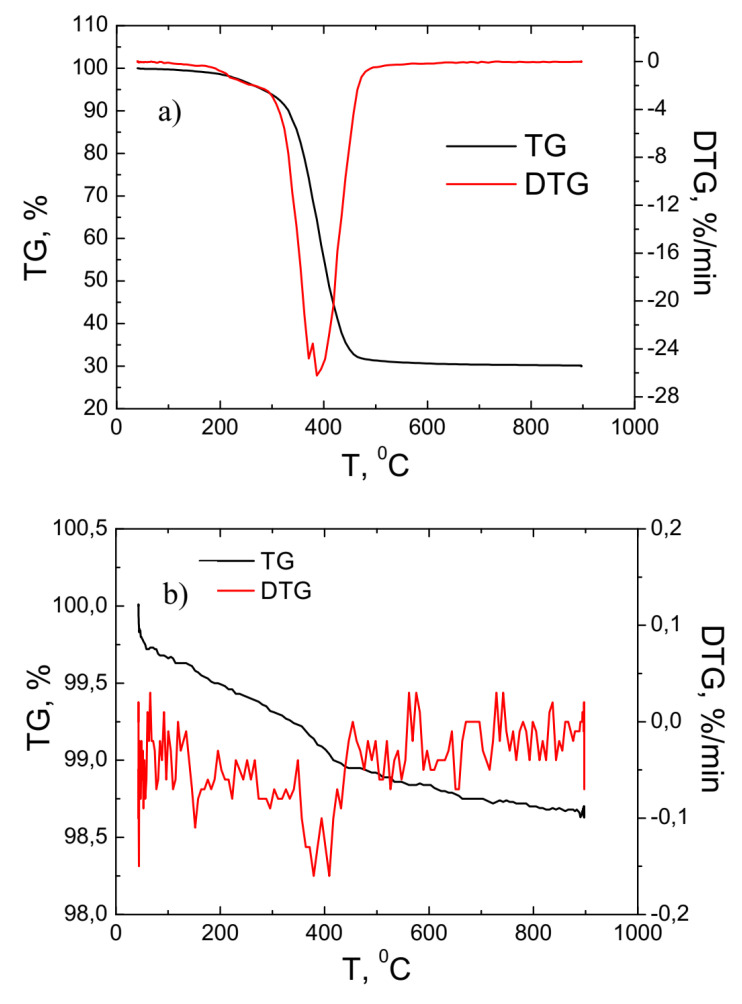
TG and DTG investigation results of the primary powder (**a**) and fiber (**b**).

**Figure 4 materials-16-03089-f004:**
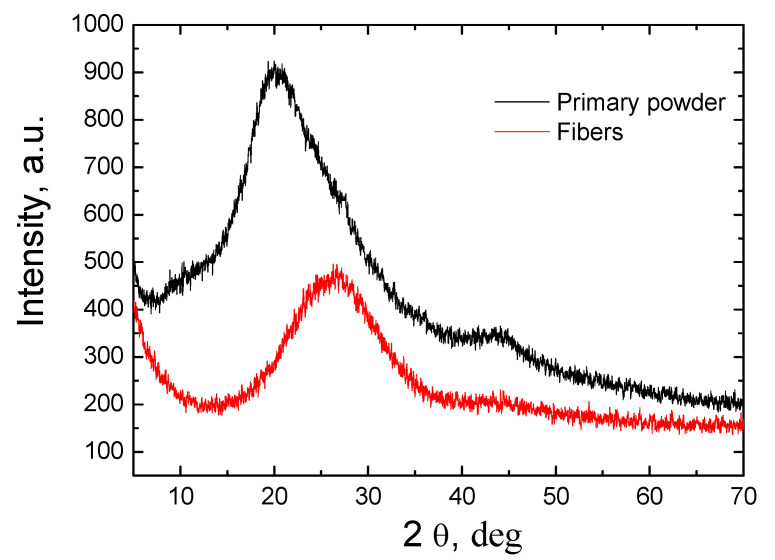
The XRD study results of the primary powder and fiber.

**Figure 5 materials-16-03089-f005:**
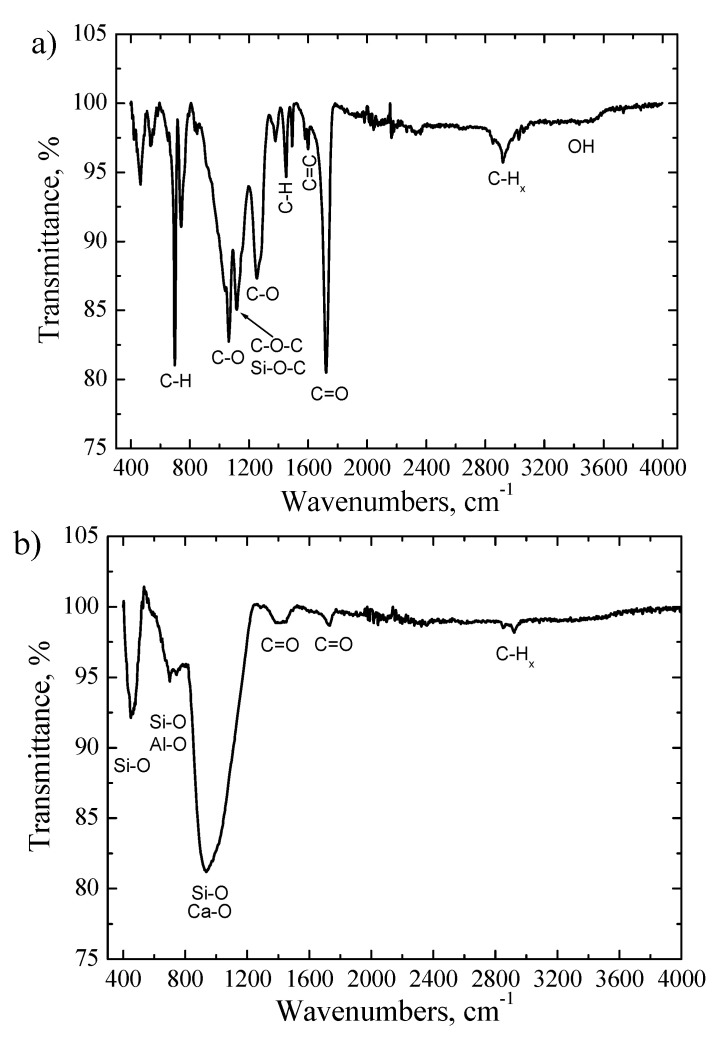
The FTIR transmittance spectra: (**a**) primary powder; (**b**) fiber.

**Figure 6 materials-16-03089-f006:**
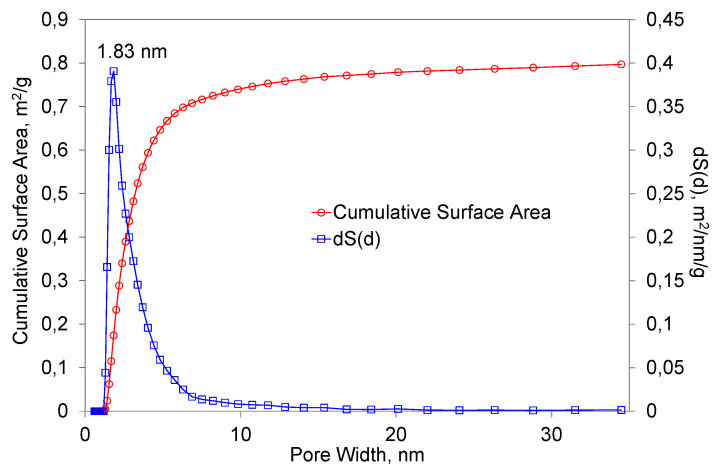
The dependence of the total surface area of the primary powders on the pore diameter.

**Figure 7 materials-16-03089-f007:**
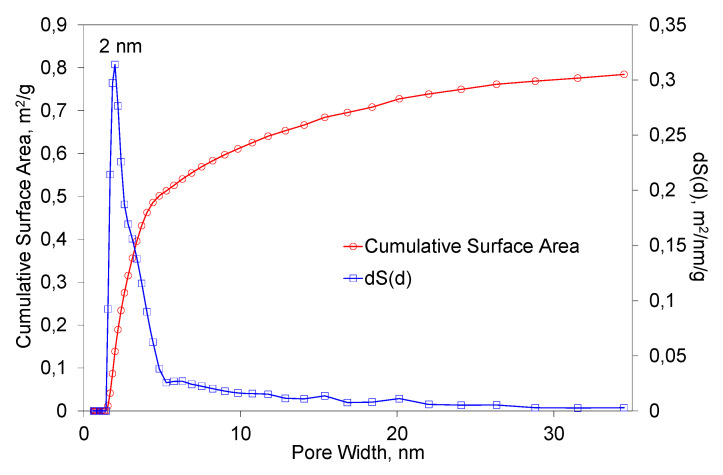
The dependence of the total surface area of the fibers on the pore diameter.

**Table 1 materials-16-03089-t001:** Results of EDS studies on primary powders and fibers.

Results of Primary Powder EDS Studies													
Units	C	O_2_	Ca	Si	Al	Na	Ti	S	Mg	F	Ni	Fe	K
**at, %**	58 ± 3	35.4 ± 1	1.5 ± 0.3	3.2 ± 0.6	1.3 ± 0.2	0.2 ± 0.03	0.1 ± 0.06	0.1 ± 0.04	0.2 ± 0.03	-	-	-	-
**wt.%**	47.6 ± 3	38.6 ± 1	4 ± 0.7	6.2 ± 1	2.3 ± 0.3	0.4 ± 0.04	0.4 ± 0.2	0.2 ± 0.1	0.3 ± 0.06	-	-	-	-
**Results of fiber EDS studies**													
**Units**	**C**	**O_2_**	**Ca**	**Si**	**Al**	**Na**	**Ti**	**S**	**Mg**	**F**	**Ni**	**Fe**	**K**
**at, %**	8.2 ± 0.5	56.5 ± 1	6.7 ± 1.3	15.5 ± 0.8	9.7 ± 2	0.4 ± 0.2	0.4 ± 0.2	0	0.7 ± 0.1	0.7 ± 0.1	0.3 ± 0.1	0.2 ± 0.1	0.1 ± 0.05
**wt.%**	4.7 ± 0.3	43.3 ± 0.8	14.3 ± 0.4	20.9 ± 1	12.6 ± 2.7	0.5 ± 0.2	1 ± 0.4	0	0.8 ± 0.1	0.6 ± 0.1	0.7 ± 0.1	0.6 ± 0.1	0.1 ± 0.05

**Table 2 materials-16-03089-t002:** The XRF study results of the primary powder and fiber.

The Results of XRF Chemical Percentage Analysis of Primary Powder												
SiO_2_	CaO,	Al_2_O_3_	TiO_2_	ZrO_2_	P_2_O_5_	Fe_2_O_3_	MgO	K_2_O	Na_2_O	BaO	SrO	SO_3_
15.4	10.7	3.38	1.7	0.5	0.34	0.34	0.33	0.17	0.15	0.15	0.12	0.11
**The results of fiber XRF chemical percentage studies**												
SiO_2_	CaO,	Al_2_O_3_	TiO_2_	ZrO_2_	P_2_O_5_	Fe_2_O_3_	MgO	K_2_O	Na_2_O	BaO	SrO	SO_3_
36.4	18.5	19.9	0.69	-	0.38	0.47	0.88	0.124	0.19	-	0.045	0.047

## Data Availability

Not applicable.

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
