# Peer review of "Recycling of Wind Turbine Blades into Microfiber Using Plasma Technology"

_materials, 2023, doi:10.3390/ma16083089_

Round 1

Reviewer 1 Report

PAPER REVIEW MATERIALS (ISSN 1996-1944)

Review of the Manuscript ID: MATERIALS 2337366 entitled “Recycling of Wind Turbine Blades Into Microfiber Using Plasma Technology” submitted to the journal of Materials.

In this paper, the authors suggest a new way of making tiny fibers from wind turbine blades that hasn't been explored before. They plan to break up the blades mechanically and turn the resulting powder into fibers using plasma techniques. The fiber production process did not produce any harmful gases and the resulting fibers were environmentally friendly. The fibers could be used in various industries, including thermal insulation, catalyst production, and construction materials.

This reviewer has some questions for the authors.

1.       Conclusion, this section is extensive. Please use a point by point conclusions of the study.

2.     What are some potential challenges in the large-scale production of these fibers, and how can they be addressed?

3.     Could this technology be applied to other types of waste materials to create useful products, and what are some examples?

4.     How can we ensure that the production of these fibers is economically viable and competitive with traditional materials?

5.      How were the micrometric fibers formed from wind turbine blades powder?

6.     What were the main peaks observed in the IR spectra of the fibers, and what caused their presence?

7.      What is the specific surface area of the primary powder and the fiber, and what does this indicate about their properties?

8.     What gas was predominant in the plasma flow used in the fiber formation process, according to the chromatographic studies?

9.     What are some of the advantages of using these fibers

10.What elements were found to be the highest percentage in the primary powder obtained by mechanical comminution of turbine blades?

11.  What type of pores dominate both the surface and the fiber?

 Reviewer suggestion: Revision is needed.

Reviewer 2 Report

This manuscript investigated the new method for recycling wind turbines using plasma technology. The abstract was well written. The introduction was well written but lacks literature about plasma technology. The authors should provide information about plasma technology and their benefits compared to other methods such as environmental impact and operating cost as well.

For the details, please kindly revise the manuscript according to the comments and suggestion below.

- Line 50: Please use a common separator number (200,000 rather than 200.000)

- Line 75: Please make sure the temperature is 5000 ºC or 500 ºC. 

- Line 79: The authors should use only one unit type as well as temperature unit in this manuscript, it should be only C or K.

- Line 102: Please provide more information about mill ML-1 such as mill type and their image that will make attraction on the reader.

- Line 107: Please kindly check the unit of gas flow rate (it should be present in same unit in line 106 that is “g/s” but in line 107 it is a velocity unit does not flow rate)

- Line 119: Please identify the aperture size or mesh number of wire screen.

- Line 154: It is better if the author revises the legend of elements in figure 2, it should be present next to figure rather than present in the figure.

- Line 221: Pleas revise the position of figure 3 on caption a) and b) and re-alignment in vertical line.

- Conclusions were well presented, but it can be written with more intensity.
